# Holography for people with no time

**Henry W. Lin[1,2], Juan Maldacena[2], Liza Rozenberg[1], Jieru Shan[1]**

[1]Jadwin Hall, Princeton University, Princeton, NJ 08540, USA

[2]Institute for Advanced Study, Princeton, NJ 08540, USA

### Abstract

We study the gravitational description of extremal supersymmetric black holes. We point out that the $AdS_2$ near horizon geometry can be used to compute interesting observables, such as correlation functions of operators. In this limit, the Hamiltonian is zero and correlation functions are time independent. We discuss some possible implications for the gravity description of black hole microstates. We also compare with numerical results in a supersymmetric version of SYK. These results can also be interpreted as providing a construction of wormholes joining two extremal black holes. This is the short version of a longer and more technical companion paper [1].

# 1 Introduction

Charged extremal black holes are very interesting objects. They seem to defy the third law of thermodynamics since they have non-zero entropy at zero temperature. In addition, their geometries develop a seemingly infinite throat with an $AdS_2 \times S^2$ (or $AdS_2 \times$(something)) geometry. These were the type of black holes for which the Bekenstein Hawking entropy was first matched with a microscopic counting [2].

It was recently understood that there is a gravitational mode whose quantum fluctuations become large as we take the low energy limit [3, 4, 5, 6]. Fortunately, we can solve exactly the quantum mechanics of this single mode [7, 8]. This modifies the naive classical gravity results at low temperatures. For the non-supersymmetric case, this implies that the density of states vanishes at low energies [9, 10, 8]. On the other hand, with $\mathcal{N} = 2$, or $\mathcal{N} = 4$, supersymmetry the density of states has a gap above zero and there is a large degeneracy at exactly zero energy above extremality [10, 11]. The $\mathcal{N} = 2$ arises for BPS black holes in $AdS_5$ [12], while the $\mathcal{N} = 4$ one describes supersymmetric black holes in $D = 4, 5$ flat space with non-zero horizon area [11].

This energy gap means that by taking a low energy limit we can clearly restrict to the ground states. In other words, we have a decoupling limit which allows us to isolate the ground states. The extremal entropy is a well studied observable in this limit, starting from [2] and including very detailed matches, as in [13].

Here we describe another set of observables which consist of correlation functions of certain operators. We consider "simple" operators that correspond to bulk fields located near the $AdS_2$ boundary, or near the region where the $AdS_2$ throat opens up into a higher dimensional spacetime. These correlators are given by Witten diagrams in $AdS_2$ which are dressed by the quantum dynamics of the boundary graviton mode, see figure 5. At low energies these correlators develop a certain universal time dependence that depends only on properties of the boundary graviton mode. For the supersymmetric case, the situation is particularly simple: they are completely time independent at very large times. These constant values depend on the details of the bulk $AdS_2$ theory.

In a dual quantum mechanical theory we can interpret these correlators as

$$\mathrm{Tr}\Big[\hat{O}_1\hat{O}_2\cdots\hat{O}_n\Big]\,, \qquad \hat{O}_i = PO_iP\,, \qquad P = \lim_{u\to\infty} e^{-uH} \tag{1}$$

where $O_i$ are the simple operators and $P$ is the projector onto zero energy states. As an example of a quantum mechanical theory with these properties we study the supersymmetric SYK model introduced in [14]. For a case with an Einstein gravity dual, we can consider the supersymmetric black hole in $AdS_5 \times S^5$ [15], whose low energy boundary gravity mode has $\mathcal{N} = 2$ supersymmetry [12], which is the case we analyze in detail. However, we expect that other supersymmetric black holes, which have a $\mathcal{N} = 4$ boundary graviton mode [11], would have similar properties.

The projector operator $P$ in (1) can be viewed as describing the zero temperature limit of the thermofield double state. More explicitly, when we write $e^{-uH}$ on the RHS of (1), we may view this as an operator acting on the Hilbert space of the one-sided quantum mechanical dual of the gravitational system, or we may view it as defining a two-sided state, e.g., the thermofield double with inverse temperature $\beta = 2u$. Taking $u \to \infty$ corresponds to the zero temperature limit of the usual two sided black hole. According to the classical solution, the length of the wormhole goes to infinity as $T \to 0$. However quantum effects kick in at temperatures of order the energy gap, and we find that its length remains finite at zero temperature. More precisely, the zero temperature state has a normalizable wavefunction which peaks at a finite length which is logarithmic in the extremal entropy, $\log S_e$.

This gives an explicit construction of a supersymmetry preserving wormhole joining two supersymmetric black holes. We can construct a large family of such wormholes by adding operators, see figure 6. We discuss how the addition of matter changes the length, making it larger. Among this family, the empty wormhole has the largest entanglement entropy, equal to the extremal entropy. The others have a smaller one.

Although the Hamiltonian is zero from the boundary[1] point of view, the bulk matter propagates and moves subject to the bulk time. So, we have a clearly emergent bulk time from a dual boundary theory with no time. In other words, an observer deep in the bulk still experiences time, despite the fact that there is no time on the boundary theory in this limit.

In this paper we summarize results from its technically heavier (super) partner [1]. We also discuss some of the conceptual implications.

This paper is organized as follows. In section 2, we consider the dynamics of the boundary graviton mode with $\mathcal{N} = 2$ supersymmetry. We discuss some implications for the description of the microstates. In section 3, we report on a numerical computation of similar correlators in $\mathcal{N} = 2$ SYK. As expected, we find agreement with the analytic computations described in the previous section, since they are governed by the effective theory. We end with further discussion in section 4.

## 2 The zero energy limit of the $\mathcal{N} = 2$ JT gravity theory

### 2.1 Density of states and the gap

By taking the extremal black hole limit, we naively expect to get a scale invariant system. But scale invariance is not compatible with the discreteness of the spectrum. One exception is if all states are precisely degenerate, so that the Hamiltonian is zero. Then the theory is not only scale invariant but also completely time independent. It becomes fully time reparametrization invariant.

In supersymmetric cases, with $\mathcal{N} = 2, 4$ supersymmetry, this is precisely what happens. One can compute the density of states using the cigar (or disk) topology and we obtain an answer with the qualitative features in figure 1 [10, 11]. There is a continuum separated by a gap from a

---

[1] By boundary, we mean the boundary of the $NAdS_2$ region, or the putative quantum mechanical dual of the region.

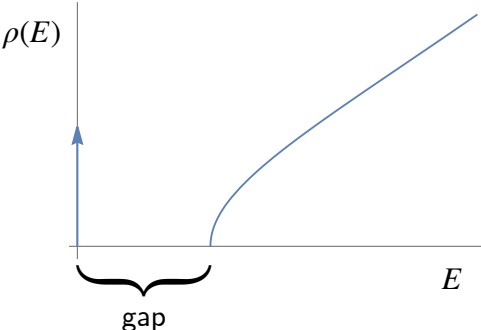

Figure 1: The density of states for the $\mathcal{N} = 2$ superSchwarzian theory in the zero R-charge sector, from [10]. We see the presence of a gap, and a delta function contribution at zero which gives the extremal entropy. $E_{\mathsf{gap}} = \frac{1}{32C}$, with $C$ defined in (2).

delta function containing a large number of degenerate zero energy states[2]. In many cases, index arguments indicate that this degeneracy should remain in the exact theory [2].

Let us comment on the energy scale that sets the gap. As we go to low energies, there is a particular gravitational mode that becomes strongly coupled. This mode can be described in terms of a reparametrization between the $AdS$ time coordinate $f$ and the asymptotic, or boundary, time $t$ [17, 5, 4, 6, 14]

$$I = -C \int \{f(t), t\} + \text{SUSY partners} \, , \qquad \text{with} \qquad \{f, t\} = \frac{f'''}{f'} - \frac{3}{2} \left( \frac{f''}{f'} \right)^2 \qquad (2)$$

where the supersymmetric partners include fermions plus a second scalar mode we will discuss later. The coefficient $C$ has units of time and its inverse sets the scale of the gap in figure 1. When we consider a charged black hole in flat space this coefficient is $C \sim S_e r_e$, where $r_e$ is the extremal radius of the black hole and $S_e$ is its extremal entropy. Notice that this is a very long time for a large black hole. $C$ also sets the time scale at which the action (2) becomes strongly coupled. We are asserting that for time scales larger than $C$, the correlators become constant, or time independent.

## 2.2 The two point functions at long distances

In this section, we sketch why the two point function has a constant value at long times

$$\langle O(u) O(0) \rangle_\beta \to \text{constant} \, , \qquad \text{for} \qquad u, \ \beta, \ \beta - u \gg 1 \qquad (3)$$

These correlation functions can be computed as follows. We can view them as the matrix element of an operator between two wormhole states, one that has been generated via euclidean time $u$ and another with time $u'$, see figure 2. We will denote such wormhole states as $|\text{TFD}(u)\rangle$. We then want to compute

$$\langle \text{TFD}(u') | e^{-\Delta \ell} | \text{TFD}(u) \rangle \qquad (4)$$

where $\Delta$ is the conformal dimension of the operator $O$. Each of these thermofield double states is a quantum superposition of wormhole states with different lengths [7, 8]. More precisely, in the

---

[2]The clear gap is a feature of the semiclassical analysis. Once we include $e^{-S_0}$ corrections, due to other topologies, we expect that the eigenvalue distribution will become smooth and there could be a non perturbatively small probability of finding an energy level in the gap region. See figure 13 in [16] for an example. We ignore such corrections here. We will show that in the SYK model for small $N$ the gap is typically clearly present.

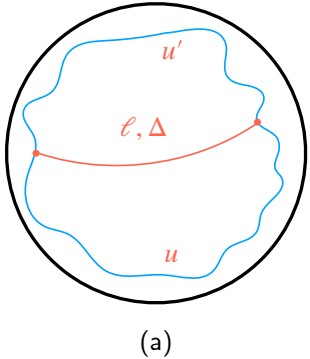
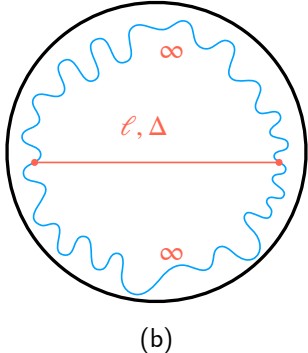

(a)                                                (b)

Figure 2: Diagrams for two point function. (a) The bottom propagator generates the thermofield double state $|\mathrm{TFD}(u)\rangle$ and the top one generates a similar one with $u \to u'$. The correlator involves a geodesic going between the two boundary points whose renormalized length is $\ell$ and the associated conformal dimension is $\Delta$. (b) The correlator in the $u = u' = \infty$ limit. The boundary has large fluctuations but the distance between the two operator insertions remains finite.

supersymmetric case, the length also has further fermionic and bosonic partners that arise from the action of the supercharges [18]. These variables are governed by a supersymmetric quantum mechanics. In the $\mathcal{N} = 2$ case this action has the form

$$I = \int du \left[ \frac{1}{4}\dot{\ell}^2 + \dot{a}^2 + i\bar{\psi}_r\dot{\psi}_r + i\bar{\psi}_l\dot{\psi}_l + \bar{\psi}_l\psi_r e^{-\ell/2-ia} + \psi_l\bar{\psi}_r e^{-\ell/2+ia} + e^{-\ell} \right] \tag{5}$$

where we have set units such that $C = 1/2$, or equivalently defined $u$ so that is related to $t$ by

$$t = 2Cu \tag{6}$$

The field $a$ can be viewed as arising from a Wilson line of a $U(1)$ gauge field in the bulk and is related to a boundary $U(1)_R$ symmetry. This field is periodic with a period $a \sim a + 2\pi$, implying that the $R$ charges are integer quantized. It is also possible to make the period larger $a \sim a + 2\pi\hat{q}$ with integer $\hat{q}$, and we will indeed consider this in section 3. To keep the discussion simple, we set $\hat{q} = 1$ for now.

The Lagrangian (5) actually has four supersymmetries. The reason is that we have two from the left boundary and two from the right boundary. Each of those two anticommutes to the same Hamiltonian. One can use these four supersymmetries to determine the form of the Lagrangian by demanding that the potential behaves like $e^{-\ell}$ at large negative $\ell$ [3].

An important feature of the Lagrangian (5) is that there are two potential terms, a repulsive potential $e^{-\ell}$ and a Yukawa term involving $e^{-\ell/2}$. We can think of the fermions as two qubits. Depending on the state of these qubits the Yukawa term could lead to a positive, zero or negative potential for $\ell$. When this $e^{-\ell/2}$ term is negative we can have bound states, see figure 3. It turns out that there is only one bound state, with exactly zero energy and also zero $R$ charge. This is due to the precise relative coefficients of the two terms in the potential, which are fixed by supersymmetry. This state preserves both the left and the right boundary supersymmetries, so it can be viewed as a supersymmetric wormhole.

---

[3]The fact that the potential agrees at large $\ell$ with the $\mathcal{N} = 0$ Schwarzian is required since the classical solutions of the $\mathcal{N} = 0$ Schwarzian are also solutions of the $\mathcal{N} = 2$ Schwarzian with $a = 0$ and all fermions set to zero. By the way, note that the classical solutions in Euclidean signature are circles in $H_2$. Since the two sides of the circle meet at a point after proper time $\beta/4$, the renormalized distance $\ell \to -\infty$ after a time $\beta/4$.

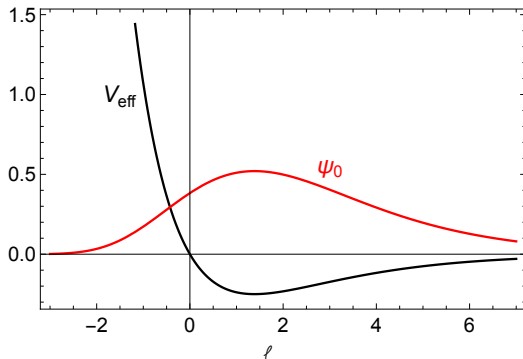

Figure 3: In black, we plot the effective potential when the term involving the fermions has an effectively negative coefficient. It leads to a single normalizable state, which is shown in red.

In this zero energy state the wavefunction has the following $\ell$ dependence, see figure 3,

$$\langle \ell | 0 \rangle \propto e^{-\ell/2} \exp\left(-2e^{-\ell/2}\right) \times (\text{fermions}) \tag{7}$$

This zero energy state, $|0\rangle$, is unit normalized and it appears in the expansion of the thermofield double as

$$|\text{TFD}(u)\rangle = e^{\frac{S_0}{2}}|0\rangle + \cdots \tag{8}$$

where the dots indicate terms that decay as $u \to \infty$. The zero energy partition function is given by

$$Z = Z(\beta = \infty) = e^{S_0} \tag{9}$$

and it sets the number of ground states. Then the long distance two point function has the form [1]

$$\lim_{u',u\to\infty} \langle \text{TFD}(u')|e^{-\Delta\ell}|\text{TFD}(u)\rangle = e^{S_0}\langle 0|e^{-\Delta\ell}|0\rangle = e^{S_0}\frac{\Gamma(1+2\Delta)}{2^{4\Delta}} \tag{10}$$

where the precise function of $\Delta$ in the right hand side requires a calculation using (7). But the fact that it is an order one number for an order one value of $\Delta$ does not require any calculation, since we have already said that $|0\rangle$ has a wavefunction localized at a value $\ell \sim 0$, (7). Furthermore, we can calculate the expectation value of $\ell$ as

$$\langle \ell \rangle = -\partial_\Delta \log\langle 2\text{pt}\rangle|_{\Delta=0} = 2(\gamma_E + \log 4) \sim o(1) \tag{11}$$

which is indeed finite and of order one. In addition, since $H = 0$, this distance does not grow in time, in contrast with the behavior at finite temperature.

In order to properly interpret these correlators (10), we need to understand how the operators were normalized. First, the distance variable $\ell$ has been defined with a subtraction from the real distance $d = \ell - 2\log\epsilon$, where $\epsilon$ is a time cutoff corresponding to the point where the $AdS_2$ joins flat space. Here both distances are in radius of $AdS_2$ units. For a Reissner Nordström black hole, $\epsilon \sim r_e/(2C) \sim \frac{1}{S_e}$.

In (10), we have normalized the operators so that their short distance expression is [1]

$$\text{Tr}\left[e^{-(\beta-u)H}Oe^{-uH}O\right] = \langle \text{TFD}(\beta-u)|e^{-\Delta\ell}|\text{TFD}(u)\rangle \sim \frac{1}{u^{2\Delta}}Z(\beta) , \qquad \text{for} \quad u \ll 1 \tag{12}$$

where the $1/u^{2\Delta}$ dependence is set by the conformal limit in $AdS_2$, which is a good approximation at relatively short distances, where the Schwarzian mode is weakly coupled. The overall coefficient in (12) sets the normalization.

It is useful to express the answer in terms of an operator $W$ whose two point function is normalized to be of order one in the boundary of $AdS_2$. More precisely we set $\langle WW \rangle \sim \left(\frac{r_e}{t}\right)^{2\Delta}$ in terms of the Schwarzschild time $t$. Recalling (6) we obtain

$$\langle \hat{W}\hat{W} \rangle_\infty = e^{S_0} \left(\frac{r_e}{2C}\right)^{2\Delta} \frac{\Gamma(1+2\Delta)}{2^{4\Delta}} \propto e^{S_0} \frac{1}{S_e^{2\Delta}} \frac{\Gamma(1+2\Delta)}{2^{4\Delta}} \tag{13}$$

The hat notation means that we have evolved over a very long Euclidean time so as to project to the ground states, as in (1).

The factor of $e^{S_0}$ disappears once we divide by the partition function (9). An interesting point about (13) is the power of $r_e/C$. This is setting the typical proper distance between the two boundaries (in units of the radius of $AdS_2$)

$$d = \ell + 2\log\frac{1}{\epsilon} \sim 2\log\frac{C}{r_e} \sim 2\log S_e \tag{14}$$

since $\ell \sim o(1)$ for the ground state (11). This should be compared to the naive classical expression

$$d \sim 2\log\frac{\beta_t}{r_e} \,, \qquad \text{classical} \tag{15}$$

which diverges[4] as $\beta_t \to \infty$. This shows that as $\beta_t$ reaches $S_e r_e \sim 1/E_{\text{gap}}$, the wormhole stops growing. Notice that we are *not* talking about the growth of the wormhole in time, but the growth as we take $\beta$ larger. It is also true that the wormhole does not grow in time, since $H = 0$ for the ground states. At zero temperature, the system is time independent and the wormhole length stays fixed at (14). Both of these features are in stark contrast with the behavior for the non-supersymmetric case. In that case the typical distance grows without bound either as $\beta \to 0$ or as time progresses. In the $\mathcal{N} = 0$ case, it has been argued that contributions from non-trivial topologies cause the distance to stop growing at values that are exponentially large in the entropy [19, 20]. Here we see see a much smaller value (14), already in the disk approximation.

## 2.3 The cylinder two point function in the probe approximation

We can also compute the two point function on the cylinder in the probe approximation[5]. Here we have one operator at one end of the cylinder and the other at the other end. We need to sum over all the states of the empty wormhole, which reduces to a sum over all the states of the Liouville-like theory (5) [21]. At low energies, we can concentrate on the zero energy state contribution which gives simply

$$\langle 2\,\text{pt} \rangle_{\text{cyl,probe}} = \langle 0|e^{-\Delta\ell}|0\rangle \tag{16}$$

which is the same as what we got for the two point function on the disk up to the number of ground states, or factor of $e^{S_0}$. We will explain this "coincidence" later (around (27), (28)).

## 2.4 $n$-point correlation functions

In the $\mathcal{N} = 0$ case, we can compute correlation functions by "dressing" the rigid $AdS_2$ correlators with propagators of the boundary particles [7]. In our $\mathcal{N} = 2$ case we can do the same. Again, when we go to long times, we can get the propagator for zero energy states. This is a function of two bulk points (and their superpartners), which we work it out in detail in [1].

---

[4]The $t$ index in $\beta_t$ is the inverse temperature in Schwarzschild time $t$, $t \sim t + \beta_t$, see (6).

[5]We are neglecting loops of particles wrapping the wormhole. This is reasonable when the typical size of the wormhole is large. We expect that this is the case for large $\Delta$. Note that these extra loops could lead to a divergence in the integration over the cylinder size, from the very thin cylinder region. We are assuming this gets cured somehow.

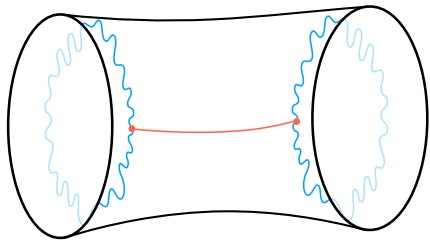

Figure 4: The cylinder diagram in the probe approximation. The states of the empty wormhole propagate along the cylinder.

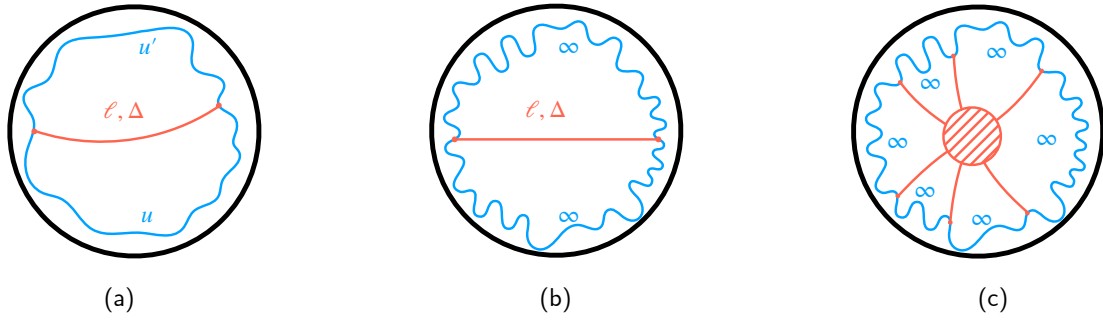

(a)                              (b)                              (c)

Figure 5: General correlators are built from $AdS$ Witten diagrams, in red, dressed by boundary graviton propagators, in blue.

The correlator is constructed from two elements. First we need the correlator of bulk fields near the boundary of $AdS_2$. We take the bulk matter to also be supersymmetric so that we may write their correlators in terms of bulk superfields. This turns out to have the form:

$$\prod_i e^{-\Delta_i \rho_i} \langle O_1(x_1, \theta_{1-}, \bar{\theta}_{2-}) \cdots O_1(x_1, \theta_{1-}, \bar{\theta}_{2-}) \rangle \tag{17}$$

where $\theta_-$ and $\bar{\theta}_-$ are Grassmann variables which implement the $\mathcal{N} = 2$ supersymmetry at the $AdS_2$ boundary. These correlators are computed by starting with the bulk fields in $AdS_2$, with no gravity, and then taking them near the boundary.

The second element is the zero energy boundary propagator $P(1,2)$ which is a function of two points whose coordinates are

$$(x, \rho, a, \theta_-, \bar{\theta}_-, \chi) \tag{18}$$

where $\chi$ is an extra Grassmann coordinate that we need to properly describe the Schwarzian degrees of freedom and $a$ keeps track of the $R$ symmetry properties. See [1] for the explicit form of the propagator; it is obtained by demanding that the appropriate supercharges of the super-Schwarzian theory vanish. Importantly, $P(1,2)$ is independent of the boundary time.

Then the final expression of any correlator at zero energies, or very long times, is given by

$$
\begin{aligned}
\langle \hat{O}_1 \cdots \hat{O}_n \rangle &= \pi e^{S_0} \int \frac{\prod_i d\mu_i}{\mathrm{Vol}(OSp(2|2))} P(1,2) P(2,3) \cdots P(n,1) \\
&\times \prod_i e^{-\Delta_i \rho_i} \langle O_1(x_1, \theta_{1-}, \bar{\theta}_{2-}) \cdots O_1(x_1, \theta_{1-}, \bar{\theta}_{2-}) \rangle
\end{aligned} \tag{19}
$$

These correlators do not depend on any boundary times since $H = 0$. But they can depend on the order, though they have a cyclic symmetry, consistent with their holographic interpretation in

(1). The $\int d\mu_i$ integrals are over the variables (18) for each point. The denominator in the measure factor arises because we are gauging the overall $OSp(2|2) = SU(1,1|1)$ symmetry of the integrand.

It turns out that the propagator contains a function of the proper distance that is the same as the wave function we already encountered in (7). This essentially implies that the propagator decays at large proper distances[6]. This implies that the correlators (19) are finite. Another feature of these correlators is that the time ordered and the out of time order correlators are of the same order, at least for low values of $\Delta$.

The time independence of (19) implies that the theory becomes "topological" at zero energies. Of course, topological in one dimension just means that the Hamiltonian is zero and the correlators are independent of time, though they can depend on the ordering. Though the theory is topological in this sense, the operators that we consider are operators that are defined by the higher energy theory. They are simple operators in the higher energy theory that are projected onto zero energy states by performing a large Euclidean time evolution on both sides, as in (1)

Note that $AdS_2$ has an SL(2) isometry group. However, the group of asymptotic symmetries is larger; it is the full group of time reparametrizations. It turns out that in the zero energy limit these are indeed symmetries of the correlators. In other words, the Schwarzian mode comes from the spontaneous breaking of this symmetry, and its action from the explicit breaking [17, 22]. This action becomes irrelevant at low energies and the integral over this mode restores the symmetry. The integral is finite thanks to supersymmetry.

## 2.5 Lorentzian continuation

These results imply that Lorentzian correlators (anchored near the $NAdS_2$ boundary) also go to a constant at long times. In particular the the two point function goes to a constant. It is important that this constant is real. This means that if one perturbs the black hole in a physical way, via a unitary process, then the effects of the perturbation will die out at long times. This is true because the change in correlation functions due to the perturbation is proportional to the commutator with the operator performing the computation. This commutator involves the imaginary part of the correlator, which vanishes at long times [1]. Notice also that the fact that the late-time Lorentzian 2-pt function is the same as the Euclidean 2-pt function implies that the bulk proper time between two points on the boundary is also finite, of order $2 \log S_e$.

If we perturb the extremal black hole by adding a simple UV operator localized in time, then we would raise its energy above extremality. We can avoid this by integrating the operator over a long Lorentzian time, a time longer than $1/E_{\text{gap}}$. This has the effect of projecting the operator to the zero energy subsector. We expect that these operators are similar to the operators $\hat{O} = POP$ that we we were discussing above in the Euclidean context. We will make this approximation when we discuss probing black holes in the next subsection.

## 2.6 Building and exploring wormholes

We have already mentioned that the state (7) corresponds to a supersymmetric wormhole of finite length. This wormhole is empty, it contains no bulk particles.

We now consider adding an operator during the Euclidean evolution that produces the wormhole state, see figure 6a. This produces a particle in the middle of the wormhole. There is a unique state

---

[6]The actual propagator depends on more than just the distance through some extra factors that, when inserted in (19), also decay at long distances.

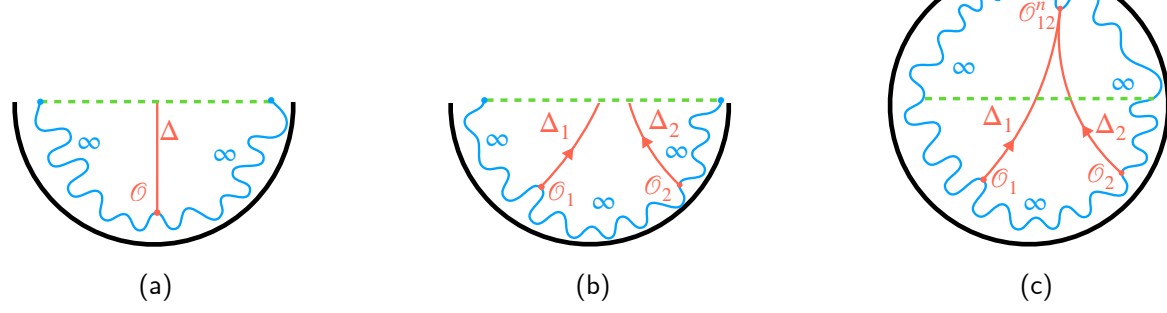

| (a) | (b) | (c) |

Figure 6: (a) State created by the insertion of an operator. It contains matter on the wormhole (indicated by green). (b) State created by the insertion of two operators, now we have two particles in the wormhole. (c) Overlap between the state in (b) and a state like (a) but created with the two particle primary operator $O_{12}^n$.

that we get by acting with a single conformal primary operator in this fashion[7]. For this state we expect that the distance between the two boundaries becomes bigger than for the empty wormhole. One way to estimate this distance is by computing an out of time order correlator involving the operator we inserted, which has dimension $\Delta$ and another operator with dimension $\Delta'$. As $\Delta' \to 0$ we are measuring the length between the two boundaries, see figure 7. Using essentially this method we find the the distance is [1]

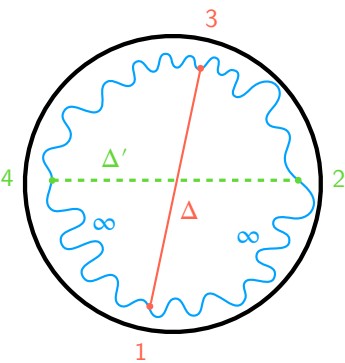

Figure 7: We consider a 4pt function involving a pair of heavy operator of dimension $\Delta \gg 1$ and a very light operator between points 2 and 4. If we do not insert any operator at 2 and 4, we can view it as giving a wormhole going between 2 and 4. This is a wormhole that contains matter in the representation with weight $\Delta$. The typical distance between 2 and 4 gives us an estimate for the size of the wormhole.

$$\ell_{24} = 2\log\Delta + o(1) , \qquad \text{for} \quad \Delta \gg 1 \tag{20}$$

which is saying that, as expected, the distance between 2 and 4 is getting larger than the empty wormhole (11).

The states we are considering would have the following interpretation in a quantum mechanical dual. The thermofield double is related to the operator $P$ that projects on to the ground states,

---

[7]By conformal primary, we mean the boundary operator that behaves as an SL(2) primary in the finite temperature conformal regime of $\text{NAdS}_2/\text{NCFT}_1$.

viewed as an entangled state. As a state on the zero energy subspace this is the identity and has maximal entropy. The state with a particle is given by

$$\hat{O} = POP \tag{21}$$

Since it is a deformation of the maximal entropy state, this state would have less entropy. The entropy can be computed via the replica trick, which involves computing $2n$ correlation functions. In general these are complicated. However, if $\Delta$ is large we expect that the OTOC contributions are suppressed, because of the increased distance we mentioned in (20). Then only planar diagrams contribute and we get an entropy that is lower than maximal by an amount that is $\Delta$ independent, for large $\Delta$ [1].

The same computation enables us to calculate the distribution of eigenvalues of the density matrix, which is simply related to the distribution of eigenvalues of $POP$. The latter turns out to be that of a gaussian random matrix, given by a semicircle law [1]. The connection between bulk fields and random matrices was made previously, and in more generality, in [23], generalizing the discussion of pure JT gravity [19].

Note that we expect that any state that contains matter will typically have lower entropy than the empty wormhole. This is particularly the case for states generated by the insertion of operators through Euclidean evolution, as we described above. The only exception would be states that are obtained by the action of a unitary operator on the low energy states. This can be achieved by performing a very slow Lorentzian evolution.

Another interesting question is the following. Imagine that now we add two particles that are well separated in the Euclidean evolution so that we get the state given by

$$\hat{O}_1 \hat{O}_2 = PO_1 PO_2 P \tag{22}$$

This is expected to be a wormhole that contains a pair of particles, see figure 6b. The two particle sate can be decomposed into a set of representations of SL(2) with dimensions $\Delta_1 + \Delta_2 + n$, which we denote as $O_{12}^n$. Each of these representations gives rise to a single wormhole state, so that the full two particle wormhole is a superposition of of wormholes each associated to a value of $n$, $PO_{12}^n P$. We can find the amplitude for each $n$ by computing the overlap, see figure 6c,

$$\langle \hat{O}_{12}^n \hat{O}_1 \hat{O}_2 \rangle \propto \mathrm{Tr}[PO_{12}^n PO_1 PO_2] \tag{23}$$

These overlaps seem to be finite and of order one, if $n$ is not large. We interpret this as saying that the two particles present in the state (22) are not very far from each other. It would be interesting to pursue this further and establish more clearly the nature of the state.

Note that the states we have discussed above, the wormhole and the wormhole plus extra matter, constitute particular entangled BPS states of two black holes, see figure 6a,b. These states are not represented by geometries that are locally supersymmetric in the interior, because the matter that we insert can break supersymmetry. However, after we include the boundary mode and project to low energies, we end up with a state that is BPS. So this is a novel way to construct BPS states and it differs from the traditional construction of BPS gravity solutions which involve finding a background with Killing spinors [24]. As an analogy, imagine we want to find a state with angular momentum zero for a rigid body. We could consider excitations on the rigid body that are not rotational invariant but we can then adjust the overall rotation of the body to produce a zero angular momentum state. Note that the two black holes that we are considering here are in different universes. If they were in the same universe, then we could wonder whether they preserve a common supersymmetry. If they are far away in flat space, they would preserve different

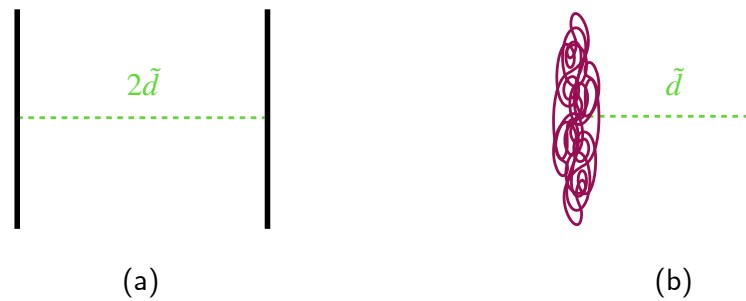

Figure 8: (a) zero energy empty wormhole has a distance $2\tilde{d}$ (26) between the two sides. (b) We claim that the results on correlation function imply that microstates that are designed to be maximally different for a given operator $O$ of order one conformal dimension should start to differ from each other only at a distance $\tilde{d}$ from the boundary.

poincare supersymmetries since they have opposite charges. It might be possible to embed them into an ambient space with suitable fluxes so that oppositely charged black holes end up being supersymmetric.

It appears that we could get an infinite amount of different states by adding various sequences of matter particles. However, we expect that the island formation phenomenon [25, 26, 27, 28] will imply that we cannot have more entanglement entropy than $2S_0$. Indeed, the finiteness of the cylinder 2-pt function is evidence of this claim, see [29]. In this paper, we restrict to a relatively low number of particles so that we do not need to worry about this.

## 2.7 Implications for the black hole microstates

In this section we make some comments on the interpretation of the long time two point function (13), which we reproduce here for convenience

$$e^{-S_0}\langle\hat{W}\hat{W}\rangle_\infty = \frac{1}{Z}\operatorname{Tr}[PWPW] = c_\Delta^2 , \qquad c_\Delta^2 = \epsilon^{2\Delta}\frac{\Gamma(1+2\Delta)}{2^{4\Delta}} \propto \frac{1}{S_e^{2\Delta}} \tag{24}$$

We discuss implications of the $c_\Delta$ factor. It sets the scale of the size of the operator in the IR compared to the operator in the $UV$. $c_\Delta$ is also telling us about the typical size of the eigenvalues of the operator $\hat{W}$

$$e^{-S_0}\langle\hat{W}\hat{W}\rangle_\infty = e^{-S_0}\sum_{i=1}^{e^{S_0}} w_i^2 \tag{25}$$

where $w_i$ are the eigenvalues of $\hat{W}$. So the typical eigenvalue is of order $c_\Delta$ (24).

It is important that they are suppressed by the factor $S_e^{-\Delta}$. This factor is present for all correlators and it arises from the propagation of the field from the boundary to a distance

$$\tilde{d} = \log S_e \tag{26}$$

into the interior, we interpret this as saying that all microstates have to agree with the $AdS_2$ geometry at least up to this distance into the interior, see figure 8.

We can call the states that diagonalize this operator, "fuzzball states", in the sense that they maximize the difference in expectation value of the operator $\hat{W}$ from state to state. We use this name because "fuzzballs" are a hypothetical representation of the black hole microstates in terms of gravity solutions, and presumably such states are designed to maximize their differences as seen

by simple gravity operators. Here we are not arguing for or against fuzzball proposals, see [30]. We are only providing constraints that those proposals should obey if we want to interpret the $AdS_2$ geometries as arising from some statistical average over such states. Since $\hat{W}$ is a gravity mode, then these results constrain the form of these gravity modes for typical solutions. Furthermore, since we expect a factor of $e^{-\Delta_i \tilde{d}}$ for each operator, this also suggests that all solutions should be similar to each other up to a distance $\tilde{d}$ from the boundary. Notice that since $\tilde{d}$ is also the distance at which it is important to consider the quantum dynamics of the boundary mode, any fuzzball proposal needs to incorporate the quantum mechanics of this mode, which is what we described in this paper. Note that the extremal black hole case is a favorable one to understand the explicit gravity description of microstates because there is no boundary time dependence.

In [1], we argue that $\hat{W}$ behaves like a random matrix with a bounded spectrum. By computing the bounds on the spectrum, one can constrain the expectation value of $\hat{W}$ in any putative fuzzball state. We expect that for $\Delta \gtrsim 1$, the maximum eigenvalue of $\hat{W}$ satisfies $|\lambda| \leq c_\Delta \times O(1)$, although we did not compute the $O(1)$ constant except in the $\Delta \gg 1$ limit where $|\lambda| \leq 2c_\Delta$.

Note that even though the vev of $\hat{W}$ in a typical eigenstate of $\hat{W}$ is $c_\Delta$, the vev of $\hat{W}$ in a general typical quantum state is much smaller. This is because a general typical state is a generic linear combination of the operators that diagonalize $\hat{O}$. In fact, we are assuming that the expectation value of $\hat{O}$ is zero, $\text{Tr}\left[\hat{O}\right] = 0$. Its expectation value could vary from state to state, but its average should obey [31]

$$\int d\mu_\psi \left[\langle\psi|\hat{O}|\psi\rangle\right]^2 = e^{-2S_0}\,\text{Tr}\left[\hat{O}^2\right] = e^{-S_0}c_\Delta^2 \tag{27}$$

which is smaller, by a factor of $e^{-S_0}$ than the normalized disk two point function in (24). In fact, this extra factor of $e^{-S_0}$ suggest that the cylinder diagram might be relevant. The cylinder two point function is supposed to arise from an average of couplings of something of the form [32]

$$e^{-2S_0}\overline{\text{Tr}\left[\hat{O}\right]\text{Tr}\left[\hat{O}\right]} = e^{-2S_0}\overline{\text{Tr}[OP]\,\text{Tr}[OP]} = \overline{\left(\frac{\text{Tr}[OP]}{Z}\right)\left(\frac{\text{Tr}[OP]}{Z}\right)} = e^{-2S_0}\langle 2\,\text{pt}\rangle_{cyl} \tag{28}$$

where we introduced factors of $e^{-S_0}$ so that we normalize the expectation values of $\hat{O}$ in the standard way. The couplings determine the form of the projector from the UV to the IR and therefore the form of the operator $\hat{O}$. There we expect that an average over couplings should have similar effects as the average over states for fixed couplings. More precisely, we have

$$e^{-2S_0}\overline{\text{Tr}\left[\hat{O}\right]\text{Tr}\left[\hat{O}\right]} = e^{-2S_0}\sum_{i,k}\int d\mu_J \,\langle i|\,\hat{O}\,|i\rangle_J \,\langle k|\,\hat{O}\,|k\rangle_J$$
$$\approx e^{-2S_0}\sum_i \int d\mu_J \left(\langle i|\,\hat{O}\,|i\rangle_J\right)^2 \approx e^{-S_0}\int d\mu_\psi \left[\langle\psi|\hat{O}|\psi\rangle\right]^2 \tag{29}$$

where $J$ denotes the couplings and $\int d\mu_J$ is the average over couplings. The IR basis elements $|i\rangle_J$ and $|k\rangle_J$ depend on the couplings. In going from the top line to the bottom line we assumed that the average over couplings would produce a $\delta_{ik}$. We then interpreted the average over couplings together with the sum, $e^{-S_0}\sum_i$, as similar to an average over states as we had in (27).

Indeed we see that (28) is the same as (27) once we use that (28) is given by the cylinder diagram (16), multiplied by $e^{-2S_0}$. This explains why we get the same value for the cylinder (16) and disk (10) two point functions, up to the expected factor of $e^{S_0}$. There is a similar relation at non-zero energies in the microcannonical ensemble, see appendix A.

The fact that the $R$ charge of the vacuum is zero has an interesting implication. It means that any operator with non-zero $R$ charge should be trivial on the BPS ground states. In fact,

for the case of black holes in flat space, modes with angular momentum on the sphere are of this kind, so all of them should be trivial on the ground states. The fact that single center BPS states carry zero angular momentum was emphasized in [33] and derived more generally from $\mathcal{N} = 4$ JT gravity in [11]. This means that any candidate "fuzzball" solution for a BPS state should be exactly spherically symmetric, and not just on average.

Motivated by the fuzzball discussion, we can ask whether there is a gravity dual of a Gaussian random state in pure $\mathcal{N} = 2$ JT gravity. Following the West Coast model [27], we conjecture that such states are dual to end of the world brane states where the brane is of order $\sim 2\tilde{d}$ from the boundary. More concretely, our conjecture is that a Gaussian random vector in a fixed $j$ charge sector is described by an end of the world brane with some tension $\mu$ and some charge $j$. To get a pure 1-sided BPS state, we project the random vector into the ground state sector using $P$. In the ground state subsector, we expect that the tension only enters in the overall normalization of the random vector[8]. We will leave a more detailed exploration of such states for the future, but let us simply remark that in such a model, there is also a simple gravity explanation of (27). In particular, the LHS of (27) is interpreted as the square of a 1-pt function, which is given by a wormhole with two end of the world branes, whereas the RHS is given by a disk computation.

From the boundary point of view, applying equation (27) gives

$$\lim_{u,u'\to\infty} \frac{\langle\psi_\mu|e^{-uH}O_\Delta e^{-uH}|\psi_\mu\rangle\langle\psi_\mu|e^{-u'H}O_\Delta e^{-u'H}|\psi_\mu\rangle}{|\langle\psi_\mu|P|\psi_\mu\rangle|^2} = e^{-2S_0}\,\mathrm{Tr}\left[\hat{O}_\Delta^2\right] \tag{30}$$

Here on the LHS we have implicitly done the disorder average over the random vector $|\psi_\mu\rangle$. Using the gravity dual (ignoring temporarily the denominator) this equation becomes:

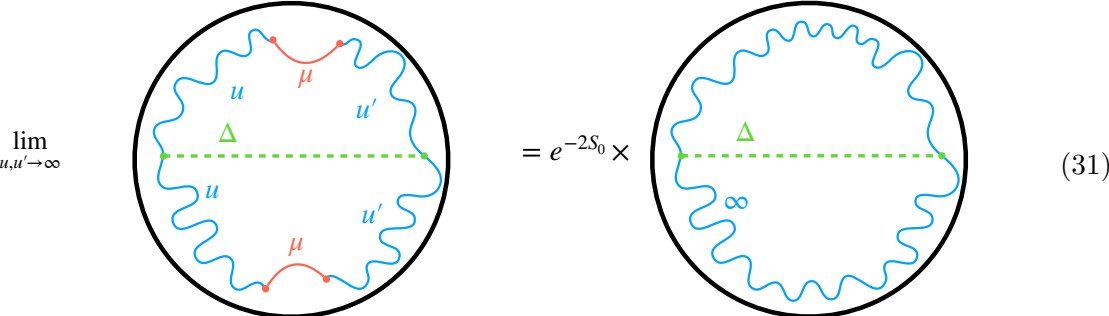

Geometrically, the above equality is saying that the end of the world brane (the red segment) is a tiny fraction of the disk when we take $u$ and $u'$ to be large. More precisely, one can compute the wormhole diagram on the LHS by using the Liouville quantum mechanics for the length mode. The end of the world brane (red) in the bottom of the diagram defines an initial state $e^{-\mu\ell}|\ell\rangle$ and the one on top defines a final state. One obtains

$$\frac{e^{S_0}\int d\ell\, d\ell'\, e^{-\mu(\ell+\ell')}\,\langle\ell'|0\rangle\,\langle 0|\,e^{-\Delta\ell}\,|0\rangle\,\langle 0|\ell\rangle}{e^{2S_0}|\int d\ell e^{-\mu\ell}\,\langle\ell|0\rangle|^2} = e^{-S_0}\,\langle 0|\,e^{-\Delta\ell}\,|0\rangle\,. \tag{32}$$

In the LHS denominator, we used the square of the disk partition function with a single end of the world brane boundary. This is in precise agreement with (27). Notice that the state defined just below the dotted green line in (31) is an empty spatial wormhole, which is described by a unique state in the Liouville description. The fact that there is an end of the world brane in the Euclidean past does not change this.

---

[8]For higher energy states, the brane should also carry two qubits worth of fermionic degrees of freedom; we will ignore these other modes.

# 3   $\mathcal{N} = 2$ supersymmetric SYK numerical results

The results we discussed above are essentially determined by the dynamics of the low energy boundary mode with $\mathcal{N} = 2$ supersymmetry. A concrete quantum mechanical model whose low energy dynamics also involves this mode is the $\mathcal{N} = 2$ version of the SYK model introduced in [14]. This is a model involving $N$ complex fermions $\psi^i$ with a random supercharge $Q = \sum_{ijk} C_{ijk} \psi^i \psi^j \psi^k$, where the $C'$s are random numbers and $\bar{Q} = Q^\dagger$. It was shown in [14] that, in the conformal regime, the dimension of $\psi$ is $\Delta = 1/6$. In addition, its $R$ charge is $1/3$. The quantum mechanics of the Schwarzian mode is slightly different than what we discussed above because $a \sim a + 2\pi \times 3$. The extra factor of 3 implies that $R$ charges come in units of $1/3$. The $R$ charge is normalized so that $Q$ has R charge one[9]. This change in the period of $a$ implies that there are actually three zero energy ground states of the theory in (5) with $R$ charges $R = \pm 1/3, 0$. So, when we compute the low energy correlators we can also specify the $R$ charge of the vacuum. In other words, the $e^{S_0}$ ground states can be separated according to these values of the $R$ charge. The analytic predictions for the low energy correlators are the same as for the case of $\mathcal{N} = 2$ JT gravity.

Here we will report on some numerical result that were obtained by performing exact diagonalization for $N = 16$.

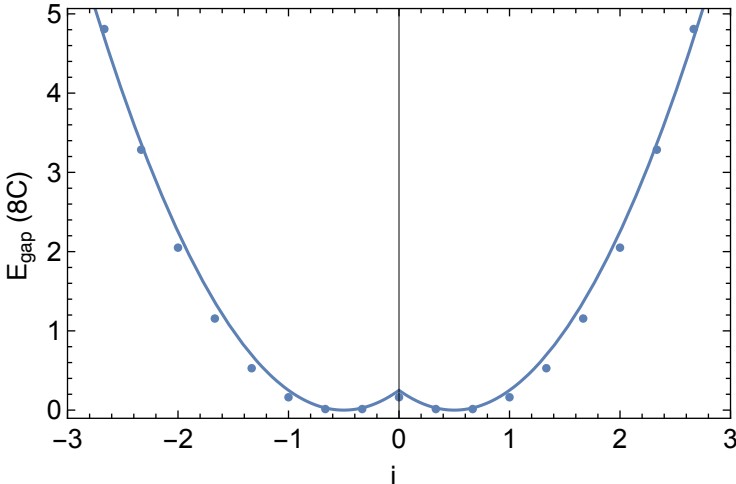

Figure 9: Gap as a function of $j$ (the $R$-charge) for a single realization of the $N = 16$ SYK model. We use the large $N$ value of the Schwarzian coupling $C = \alpha_s N = 0.1347$. The solid curve is the analytic prediction, see equation (33).

As a first question, we wanted to confirm the gap predicted by [10] for states with $R$ charge $j$

$$E = \frac{1}{8C} \left( |j| - \frac{1}{2} \right)^2 , \qquad C = \alpha_S J N , \qquad \alpha_s = 0.00842... \tag{33}$$

where we have given the large $N$ expression for $C$ as well as the value for $\alpha_S$ computed by solving the large $N$ Schwinger-Dyson equations in [34]. The gaps are plotted in figure (9) as a function of the $R$ charge.

As a next question, we can calculate the value of various operators on the BPS ground states

---

[9]We will also assume $N$ is even, since for $N$ odd there is a further necessary modification that we will not discuss here [10].

| Operator | Vacuum $R$-charge | Schwarzian prediction | $N{=}16$ SYK | Error |
|:---:|:---:|:---:|:---:|:---:|
| $\psi_i$ | 0 | 0.111 | $0.110 \pm 0.005$ | $-1\%$ |
| | $-1/3$ | 0.111 | $0.110 \pm 0.005$ | $-1\%$ |
| $\psi_i\psi_j$ | $-1/3$ | 0.0247 | $0.024 \pm 0.003$ | $-4\%$ |
| $\bar\psi_i\psi_j$ | $-1/3$ | 0.0282 | $0.027 \pm 0.001$ | $-4\%$ |
| | 0 | 0.0874 | $0.079 \pm 0.001$ | $-9\%$ |
| | $+1/3$ | 0.0282 | $0.027 \pm 0.001$ | $-4\%$ |

Table 1: Comparison between exact diagonalization results for the 2-pt function with the Schwarzian predictions. Here we use the large $N$ value for the Schwarzian, $C = 0.00842N = 0.135$. The $R$-charge corresponds to the $R$-charge of the set of zero energy states, see equation (34). The error bar we display in the fourth column is the standard deviation obtained by changing the value of $i$, the index of the operator. This is related to the error we would get if we vary the coupling constants.

with various charges. In other words, for various operators $O$ we compute

$$\frac{1}{N_{\mathsf{TFD}}}\,\mathrm{Tr}\Big[O_r^\dagger P_{j+r}O_r P_j\Big]\;, \qquad N_{\mathsf{TFD}} = \mathrm{Tr}[P]\;, \qquad P = \sum_j \mathrm{Tr}[P_j] \tag{34}$$

where $P_j$ is the projector onto zero energy states with $R$ charge $j$, and $r$ is the $R$ charge of the operator $O$. As we mentioned above, the possible values of $j$ for the zero energy states are $j = 0, \pm 1/3$, a fact that we have also confirmed numerically. We have numerically computed (34) for $N = 16$. We have also analytically computed the expected large $N$ answer based on the formula (24), after taking into account the proper UV normalization of the operators, and set $N = 16$ in that result[1]. The comparison between these two computations is displayed in table 1.

As another comparison, we can compute the OTOC vs the OTOC for the basic fermions. In other words, we can compare

$$\mathsf{TOC} = \mathrm{Tr}\big[P\bar\psi^j P\bar\psi^i P\psi^i P\psi^j\big]\;, \qquad vs \qquad \mathsf{OTOC} = -\,\mathrm{Tr}\big[P\bar\psi^i P\bar\psi^j P\psi^i P\psi^j\big] \tag{35}$$

It turns out that the two expressions are identical, due to supersymmetry, as we explain in [1]. However, this illustrates the point that the two correlators can be similar. The fact that they are identical is a special feature of these operators, which are BPS, but it is not true for non BPS operators. In fact, considering neutral operators of the form $O_1 = \psi^i\bar\psi^j$ and $O_2 = \psi_k\bar\psi_l$, together with their adjoints, we numerically find a ratio of the OTOC to the TOC of 76%.

We have also found the eigenvalue distribution of $\psi^i\bar\psi^j$ operators and found agreement with random matrix expectations [1], which are modified by the low value of $N$ ($N = 16$).

# 4 Discussion

We have described some aspects of the zero energy limit of $\mathcal{N} = 2$ supersymmetric black holes. We need supersymmetry so that we get a finite number of states at exactly zero energy. The presence of an energy gap helps us to argue that we can effectively project on these states by evolving the system by a sufficiently long amount of euclidean time evolution. This amount is of order the black hole entropy $S_e$, rather than the exponential of the entropy. After taking this limit the Hamiltonian of the system is zero, and we effectively have no boundary time. In this limit the SL(2) symmetry of $AdS_2$ is enhanced to its full asymptotic symmetry: time reparametrizations.

Note that the final expression for the correlators, (19) is a quantity we can calculate purely in $AdS_2$. So we have a correspondence between $AdS_2$ and a topological quantum mechanics, which is simply a set of $e^{S_0}$ states and the corresponding operators. However, we seem to have a preferred choice of operators which are the operators that are simple in the bulk. This choice of operators is natural in the higher energy theory with boundary time, but there is no obvious reason to choose these operators from the purely IR boundary theory. We have mainly considered the disk topology (specially in (19)) and it would be interesting to consider the effect of higher topologies in this zero energy sector. Of course, for the non-zero energy sector these effects were the subject of a number of interesting recent papers including, among others, [19, 20, 35, 29, 36]. At the disk level, we have an apparently infinite choice of operators, since we can have multiple field insertions. These loop corrections should limit the set of operators to the finite set of $e^{S_0} \times e^{S_0}$ matrices. A toy model where higher topologies lead to a finite number of states was given in [37].

It was argued in the past that we could not have a dual to a purely $AdS_2$ gravity theory [38, 39]. This construction evades these arguments because it breaks the link between the time coordinate in the $AdS_2$ bulk (and its associated energy) and the boundary time or boundary energy.

At the disk level, there is a bulk symmetry operator which is the matter Casimir. This is a combination of the matter bulk generators defined in [40, 41] which commutes with the Hamiltonian and therefore survives the low energy limit[10]. It would be interesting to identify this generator in the quantum mechanics theory, though perhaps it is only defined in its $S_0 \to \infty$ limit. This symmetry generator can be used to identify the lightest simple bulk operators.

The correlation functions (19) are still non-trivial and they depend on some features of the bulk theory such as the masses and couplings in the bulk. The two point functions are relatively easy to calculate and we have compared these predictions against numerical SYK computations. They were found to agree surprisingly well. We can view these as numerical checks of some quantum aspects of supergravity theories.

Though we focused on the $\mathcal{N} = 2$ supersymmetric case, we expect similar results for the $\mathcal{N} = 4$ case. In fact, the qualitative fact that there is a disconnect between bulk time and boundary time is also present for the $\mathcal{N} = 0$ case, except that in that case the theory develops a universal dependence on boundary time, as we discuss in more detail in [1].

We have considered here mainly the disk diagram. We do not expect large modifications from the presence of higher topologies, for the simple observables we have discussed here. The disk expectation values are all small, but they are suppressed by inverse powers of $S_e$, while corrections from other topologies is expected to involve powers of $e^{-S_e}$.

It is generally expected that black holes should be associated to chaotic systems. Since the Hamiltonian is zero, we could wonder where the chaos is in our case[11]. The idea is that the projection operator $P$ on the low energy sector should be chaotic in some sense. In particular, one expects that the eigenvalues of the operator $\hat{O}$ would display random matrix statistics. For operators with large dimensions we show in [1] that we get a semicircle law, as for gaussian random matrices. But we did not check for the distribution of pairs of eigenvalues. The fact that matter fields in JT gravity should be viewed as random matrices was previously discussed in [23] for general JT gravity theories, and the discussion here is just a limit of that general analysis.

It would be desirable to have a better Lorentzian understanding of this system. In particular, one would like to have an understanding of possible bulk singularities. Based on the Euclidean computations we would expect that a bulk observer has a peaceful existence at least up to a time of order the radius of $AdS_2$, since the $t = 0$ Cauchy slice seems perfectly reasonable.

---

[10]The discussion in [40, 41] was for the $\mathcal{N} = 0$ case, but we expect a similar story for $\mathcal{N} = 2$.
[11]This question was raised by S. Shenker.

There are some vague structural similarities to the case of de-Sitter space. In both cases there is a Hamiltonian that is zero. In both cases there is a simple state where the entanglement entropy is maximal and the addition of matter can only lower it. In both cases time emerges from a system with no time. We are not saying that this is a model for de-Sitter. But we are saying that understanding how time emerges in this case, where we do have a candidate quantum mechanical dual prescription, might help us understand the de-Sitter case where there is no clear quantum mechanical dual.

### Acknowledgments

We would like to thank Daniel Jafferis, Henry Maxfield, Baurzhan Mukhametzhanov, Vladimir Narovlansky, Geoffrey Penington, Douglas Stanford, Phil Saad, Stephen Shenker and Gustavo Turiaci for comments and discussion.

J.M. is supported in part by U.S. Department of Energy grant DE-SC0009988 and by the Simons Foundation grant 385600.

## A   Relation between the disk and the trumpet at non zero energy.

We have seen that the disk and cylinder diagrams were related when we take the zero energy limit. Here we point out that they are also related at at non-zero energy as long as we restrict to a very small energy window, this follows simply from observations in [21, 29].

Then the disk can be viewed as

$$D_2 = e^{S_0}(\rho_E \delta E)^2 \langle E|e^{-\Delta \ell}|E\rangle \tag{36}$$

where I approximated $E' \sim E$ in the matrix element since we assume that the energy window, $\delta E$ is very small, $\delta E \ll 1$. The cylinder diagram is is

$$T_2 = \rho_E \delta E \langle E| e^{-\Delta \ell} |E\rangle \tag{37}$$

Then if we define $d_E = e^{S_0}\rho_E \delta E$, then we see that the two computations are related as expected from the equality between the average over states and the average over couplings discussed around (27) (28).

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
