# Peer review of "Holography for people with no time"

_SciPost Physics_

## Round 2 · Referee Report · Anonymous (Referee 2) · 2022-11-21

Strengths

Overall a nice summary (companion) paper with interesting results.

Weaknesses

Sometimes a little hard to understand without looking at the longer paper at the same time.

Report

The authors study correlation functions in extremal black holes with N=2 supersymmetry, using the exact description of the AdS2 near horizon geometry. In the zero energy limit, quantum gravity effects are large and a finite number of degenerate ground states separated from the continuum by a gap, lead to two-point functions with a constant large time limit. The boundary description in this limit is `topological' in the sense that the Hamiltonian disappears and the full time reparametrization symmetry is restored. Various implications on topics of current interest are discussed.

The results are very interesting and timely. The paper is often quite schematic. I think this is ok in this particular case because it is a companion paper to a longer paper, which contains more details. The question is then whether this companion paper deserves to be published separately. I believe the answer is definitely yes, as this manuscript does offer a nice summary of the results and presents them in a useful way. Also, some of the discussion (such as microstates) is more detailed here than in the long paper.

Sometimes the paper is a bit hard to follow, which is not the desired purpose of a companion paper. E.g., section 2.7 (which contains content that's not in the long paper) could perhaps be expanded a bit to make it more self-contained (e.g. the "West Coast model" is mentioned without further explanation or reference). Similarly, the discussion of correlation functions around (17)-(19) is perhaps a bit too schematic and hard to follow without consulting the longer paper. It would be good if this could be expanded a bit.

---

## Round 2 · Referee Report · Anonymous (Referee 1) · 2022-11-21

Strengths

Good paper exploring some important implications of quantum effects in supersymmetric black holes.

Weaknesses

Relies on companion paper for details, some aspects are hard to comprehend.

Report

The paper explores an interesting class of observables in supersymmetric black holes by focusing attention on the near-horizon region. The authors argue that natural observables are correlation function of operators inserted on the boundary of the near horizon region, which by virtue of the near-AdS2 asymptotics and focus on the low-energy or long-time limit, have time-independent (constant) correlations. The paper is a summary of a companion paper and as such avoids the technical details. However, certain aspects of the discussion are somewhat hard to comprehend in the present manuscript. While I think the paper is of high-quality and should be accepted for publication, I would like to encourage the authors to consider making it a bit more self-contained. Some specific issues are highlighted below:

1. The thermofield double state used in the discussion could do with some clarification. Prior to taking the near-horizon limit, the states are BPS (sticking to supersymmetric black holes), and one is usually led to consider the microcanonical ensemble of states in the BPS subspace of the Hilbert space. While I understand the thermofield double state to be the state obtained from the global AdS2 with its two boundaries, explaining how this is related to the UV description would be helpful. Relatedly, should the Hamiltonian specified in (1) which vanishes be seen as the BPS Hamiltonian of the UV description? While the authors are clear, I think it would also would help the casual reader to be aware that `boundary' always refers to the near-horizon boundary.
2. A minor pedantic comment: Eq (7) as written is a bit strange. The l.h.s is a vector in Hilbert space, while the r.h.s appears to be a representation in a particular basis.
3. In Section 2.5 are the Lorentzian correlations still restricted to the boundary of the AdS2 region? Is the statement made here equivalent to saying that there is no response from the near-horizon region. What correlations are captured by operators that are smeared over a timescale of order $1/E_{gap}$?
4. The discussion of composite operators in Section 2.6 is mildly confusing. In usual AdS CFT dictionary bulk fields correspond to primaries of the CFT. Descendants or multiparticle states are excitations of the same bulk field. Does this picture hold for the AdS2 region under discussion or is it modified to include new operators corresponding to multiparticle excitations?
5. Finally, it would be useful to have some brief comments on how the operators discussed in the near-horizon region should be viewed from the UV perspective.

---

## Editorial Decision

resubmitted